# Differential Upregulation of Th1/Th17-Associated Proteins and PD-L1 in Granulomatous Mycosis Fungoides

**DOI:** 10.3390/cells13050419

**Published:** 2024-02-27

**Authors:** Mario L. Marques-Piubelli, Jesus Navarrete, Debora A. Ledesma, Courtney W. Hudgens, Rossana N. Lazcano, Ali Alani, Auris Huen, Madeleine Duvic, Priyadharsini Nagarajan, Phyu P. Aung, Ignacio I. Wistuba, Jonathan L. Curry, Roberto N. Miranda, Carlos A. Torres-Cabala

**Affiliations:** 1Department of Translational Molecular Pathology, The University of Texas MD Anderson Cancer Center, Houston, TX 77030, USAdaledesma@mdanderson.org (D.A.L.); jlcurry@mdanderson.org (J.L.C.); 2Department of Anatomical Pathology, The University of Texas MD Anderson Cancer Center, Houston, TX 77030, USA; jenasuv1985@gmail.com (J.N.);; 3Department of Dermatology, The University of Texas MD Anderson Cancer Center, Houston, TX 77030, USA; 4Department of Hematopathology, The University of Texas MD Anderson Cancer Center, Houston, TX 77030, USA

**Keywords:** T helper, Th17, Th1, granulomatous mycosis fungoides

## Abstract

Granulomatous Mycosis Fungoides (GMF) is a rare form of mycosis fungoides (MF) characterized by a granulomatous infiltrate associated with the neoplastic lymphoid population and is considered to have a worse prognosis compared with regular MF. The upregulation of the T helper (Th) axis, especially Th17, plays an important role in the pathogenesis of several inflammatory/infectious granulomatous cutaneous diseases, but its role in GMF is still not elucidated to date. In this study, we evaluated the immunohistochemical expression of Th1 (Tbet), Th2 (GATA-3), Th17 (RORγT), T regulatory (Foxp3), and immune checkpoint (IC) (PD-1 and PD-L1) markers in a cohort of patients with GMF and MF with large cell transformation (MFLCT). Skin biopsies from 49 patients (28 GMF and 21 MFLCT) were studied. Patients with GMF were associated with early clinical stage (*p* = 0.036) and lower levels of lactate dehydrogenase (*p* = 0.042). An increased percentage of cells positive for Tbet (*p* = 0.017), RORγT (*p* = 0.001), and PD-L1 (*p* = 0.011) was also observed among the GMF specimens, while a stronger PD-1 intensity was detected in cases of MFLCT. In this cohort, LCT, RORγT < 10%, Foxp3 < 10%, age, and advanced stage were associated with worse overall survival (OS) in univariate analysis. GMF demonstrated Th1 (cellular response) and Th17 (autoimmunity) phenotype, seen in early MF and granulomatous processes, respectively, which may be related to the histopathological appearance and biological behavior of GMF. Further studies involving larger series of cases and more sensitive techniques are warranted.

## 1. Introduction

Granulomatous formation can occur in up to 2% of all cutaneous lymphomas, and it is reported in a broad variety of them [1,2]. Granulomatous mycosis fungoides (GMF) is an unusual variant of mycosis fungoides (MF) and represents the most common form of granulomatous primary cutaneous T-cell lymphoma [1]. Clinical presentation is usually similar to classic mycosis fungoides (MF) or other benign inflammatory granulomatous diseases and is histologically composed of an epidermotropic and diffuse infiltrate of atypical lymphocytes associated with granulomatous inflammation [3,4]. Large cell transformation (LCT) is characterized by the transformation of small neoplastic lymphocytes to large and clonally identical forms and is accepted as a worse prognosis factor [5]. Although both GMF and MF appear to have a similar rate of LCT, GMF appears to portend a worse prognosis [2,5]. Disease progression in GMF has been reported to be as high as 46%, in contrast to 30% in classic MF, and is comparable to that reported for MF with LCT [6,7].

The T helper (Th) is a subset of CD4+ T-cells of the adaptive immunity response and, along with the T regulatory (Treg) component, they help in the activation of the B- and cytotoxic T-cells counterparts [8]. Based on distinct cytokine profiling and specific effects on the immune system, several types of Th response are recognized, such as T helper 1 (Th1), T helper 2 (Th2), and T helper 17 (Th17). While the upregulation of the Th axis appears to play a role in the pathogenesis of infectious and inflammatory cutaneous diseases and systemic forms of T-cell lymphomas, our understanding at a protein level of its role in different types of primary cutaneous T-cell lymphomas is very limited [9,10,11]. The cytokine profiling of the Th17 response of the immune microenvironment in cutaneous T-cell lymphomas has been assessed, and it seems to be dysregulated, promoting an imbalance of the neutrophil component [12]. 

Our knowledge about the biology, epidemiology, and clinicopathologic features of GMF is currently restricted to information obtained from a few case series and reports [1,2,7,13,14,15,16]. Importantly, the composition of the immune checkpoint (IC) landscape and status of other biomarkers that are expressed by the neoplastic lymphocytes in GMF and that could be implicated in its reported different behavior is unknown. Hence, the identification of the relevant biomarkers used in clinical practice that could translate in successful therapies is a high priority. In this study, we aimed to describe the clinicopathologic characteristics of GMF in a single tertiary cancer center and evaluate the expression of IC and Th proteins in this disease, comparing them with a known aggressive form of MF, MFLCT. 

## 2. Materials and Methods

### 2.1. Patient Selection

This study was approved by the Institutional Review Board of the University of Texas MD Anderson Cancer Center and was conducted in accordance with the principles of the Declaration of Helsinki. This retrospective cohort included patients with a diagnosis of GMF or MFLCT between 01/2003 and 12/2017, for whom available formalin-fixed embedded (FFPE) skin biopsies were used for correlative analysis. The original diagnosis was confirmed by two dermatopathologists (CAT-C and JLC). GMF was defined as a prominent granulomatous infiltrate associated with the malignant lymphoid infiltrate without the typical clinical features of granulomatous slack skin, while MF with LCT (MFLCT) denoted the histopathological presence of large atypical mononuclear cells representing more than 25% of the infiltrate (Figure 1). Clinicopathologic variables were collected from electronic medical records and included date of diagnosis, age, sex, ethnicity, personal oncologic history, lactate dehydrogenase (LDH) levels, β-2 microglobulin (β2M) levels, lesion type, pathologic diagnosis, clinical stage at the time of diagnosis (Table 1), and date of last follow-up and status.

### 2.2. Immunohistochemistry (IHC) Analysis

After routine diagnostic assessment, 4 μm thick sections of formalin-fixed paraffin-embedded skin biopsies were used for immunohistochemical staining to assess the expression of proteins related to Th response. The reaction was performed on a Leica Bond RXm autostainer (modified version of the standard Leica “F” protocol) and the process included deparaffinization, rehydration, pretreatment in 0.01 M sodium citrate buffer (pH 6.0) for 10 min at 95 °C, and incubation with 1% hydrogen peroxidase. The sections were incubated with 0.01M phosphate-buffered saline and incubated overnight with antibodies following specific conditions: TBX21 (D6B8B, Cell Signaling #13232, Danvers, MA, USA, 1:100), GATA3 (D13C9, Cell Signaling #5852, 1:100), RORγT (6F3.1, EMD Millipore #MABF81, Burlington, MA, USA, 1:800), Foxp3 (206D, BioLegend #320102, San Diego, CA, USA, 1:50), PD-1 (EPR4877-2, Abcam #ab137132, Cambridge, UK, 1:250), and PD-L1 (E1L3N, Cell Signaling #13684, 1:100). The sections were then washed and incubated with goat antirabbit IgG biotinylated secondary antibodies and counterstained with hematoxylin. Tonsils and reactive lymph nodes were used as positive controls. The assessment of each marker was performed using standard microscopy, and the atypical lymphoid infiltrate in each biopsy was scored by two pathologists (MLM-P and CAT-T) in 10% increments (0 to 100%). PD-L1 labeling was considered positive when a membranous pattern was present. The intensity of each marker was graded as absent (0+), mild (1+), moderate (2+), and intense (3+) based on the predominant pattern in the case. Information regarding the additional markers used in the diagnostic assessment is summarized in the Appendix A (Appendix A). 

### 2.3. Statistical Analysis

Association between categorical variables was performed using χ^2^ or Fisher’s exact tests, while the Mann–Whitney test or an ANOVA were used for continuous variables, as appropriate. Overall survival (OS) was defined as time from the diagnosis to death or last follow-up and calculated for all patients in the study using the Kaplan–Meier estimate. The different subgroups were compared using log-rank test and the multivariate analysis was not performed due the sample size. The number of events per variable is summarized in Appendix A. A *p*-value of ≤0.05 (two-tailed) was considered statistically significant (95% confidence interval [CI]). Statistical analyses were performed using IBM SPSS Statistics for Windows, version 24 (IBM Corp., Armonk, NY, USA). 

## 3. Results

### 3.1. Patient Baseline Characteristics

Forty-nine patients were included in the study and divided into two main groups: GMF (n = 28) and MFLCT (n = 21). All the clinicopathologic variables and immunohistochemical positivity rates of the markers used in the diagnostic assessment are summarized in Table 1 and Table 2. 

### 3.2. GMF Group

The median age at presentation in this group was 58 years (range: 21–67 years), and most patients were male (54%) and White (68%). Six patients (21%) had a previous history of malignancy, including classic Hodgkin lymphoma (CHL) (n = 1), meningioma (n = 1), papillary thyroid carcinoma (n = 1), nonspecified skin carcinoma (n = 1), breast carcinoma (n = 1), and renal cell carcinoma, prostate carcinoma, and squamous cell carcinoma of the skin (n = 1). Around 90% of patients had a previous treatment for GMF at the time of the analyzed biopsy, while 100% of patients with MFLCT had a previous history of treatment. The median LDH at initial clinical presentation was 501 UI/L (range: 128–836 UI/L), while the β2M was 2.45 mcg/mL (range: 2.4–4.8 mcg/mL). Patches and/or plaques were the most common lesion types, and patients presented predominantly at Stage I (39%) and II (29%) of disease. The histopathological features of LCT were detected in only two cases (2/28). IHC studies (Table 2) showed the neoplastic infiltrate to be composed of CD3-positive (24/24, 100%) and CD4-positive (22/23, 95%) T-cells. The expression of CD30 was frequent (17/19, 89%, median expression: 4%). T-cell receptor (TCR) gene rearrangement was performed in four cases and had the following results: monoclonal beta and gamma (2/4, 50%), monoclonal gamma (1/4, 25%), and oligoclonal beta (1/4, 25%). The median follow-up (FU) was 77 months (range: 11–279 months), and the median OS was 132 months (95%CI: 89.9–174.1 months). Fifteen patients (54%) were alive with disease (AWD), twelve (42%) died of disease (DOD), and one (4%) was alive with no evidence of disease (ANED) at the last FU. 

### 3.3. MFLCT Group

The control group represented by MFLCT had a median age of 61 years (range: 34–77 years). These patients were predominantly males (67%), White (76%), and all were previously treated at the moment of the current biopsy. Five patients (23%) had a previous history of malignancy, as follows: nonspecified skin carcinoma (n = 1), CHL (n = 1), squamous cell carcinoma of the skin (n = 1), cutaneous melanoma (n = 1), and prostate carcinoma and nonspecified skin cancer (n = 1). The median LDH was 583 IU/L (range: 338–1035 IU/L), and the median β2M was 3.15 mcg/mL (range: 1.6–4.5 mcg/mL). Patches and plaques were the most common lesion types, and most patients had Stage II (38%) of disease. IHC studies (Table 2) showed the lymphoma cells to be positive for CD3 (8/8, 100%), CD4 (7/8, 87.5%), CD7 (2/3, 66%), and CD30 (20/21, 95%, median expression: 80%). TCR gene rearrangement was performed in three cases, with monoclonal gamma in two cases (2/3, 67%) and monoclonal beta in one case (1/3, 33%). The median FU in this group was 70 months (range: 2–250 months), and the median OS was 70 months (95%CI: 48–91.9 months). Eighteen patients (86%) died of disease, two patients (9%) were alive with disease, and one patient (5%) was alive with no evidence of disease at the last FU. 

### 3.4. Comparison of Baseline Characteristics between GMF and MFLCT

The GMF group had a statistically significant association with lower levels of LDH at clinical presentation (median, 501 vs. 583 IU/L) (*p* = 0.042) and lower clinical stage (Stage I) at the time of diagnosis (*p* = 0.036) when compared to MFLCT. While the total number of positive cases for CD30 (≥1% positivity in the neoplastic cells) was not statistically significant, cases of MFLCT had a higher median percentage of expression of this marker when compared to cases of GMF (median%: 4% vs. 80%). Other clinicopathologic variables such as age, sex, ethnicity, previous history of malignancy, history of previous treatments for MF, β2M levels, clinical presentation, immunophenotype, and TCR gene rearrangement had no statistically significant differences between the compared groups. 

### 3.5. Upregulation of Th1/Th17 and ICI Markers in GFM

The median and mean (SD) expressions of the Th markers are summarized in Table 3, and the first measure was used as a cut-off for the correlation with outcomes. The GMF group was associated with a statistically significant higher expression of Tbet (median, 20% vs. 0%), RORγT (median, 15% vs. 0%), and PD-L1 (median, 10% vs. 0%) when compared to MFLCT (Figure 2). The expression of GATA3 (median, 40% vs. 40%), Foxp3 (median, 15% vs. 10%), and PD-1 (median 10% vs. 10%) had no association with the groups. No association between staging and the expression of Tbet, RORγT, or PD-L1 was identified.

Among positive cases, the intensity of the Th markers in both groups was predominantly distributed in the 3+ category (Table 3). PD-1 with 3+ intensity was significantly more common in the MFLCT group when compared to GMF (Figure 3). Of note, no cases of predominant 1+ expression were found in any of the two groups. 

### 3.6. Staging and Correlation with Outcomes

For all patients in this series, the median follow-up was 76 months (range: 2–279 months), and the median OS was 121.6 months (95%CI: 94.1–149.1 months). Table 4 summarizes the results of the univariate (UVA) analysis. The following factors were significantly associated with worse OS on UVA: patient’s age (*p* = 0.017), clinical stage IV (*p* < 0.001), MFLCT group (*p* = 0.018), RORγT positivity <10% in the neoplastic cells (*p* = 0.006), and Foxp3 positivity <10% in the neoplastic cells (*p* = 0.002). Appendix A summarizes the number of events per variable. 

## 4. Discussion

GMF represents an uncommon variant of cutaneous T-cell lymphoma (up to 6% of all MF cases) and is considered to portend a worse prognosis [7,17,18]. The progression-free survival (PFS) of GMF has been reported as 59% and 33% at 5 and 10 years compared to 84% and 56% for classic MF [7]. The PFS of GMF is closer to that shown by MFLCT (45% and 22%), making MFLCT a reasonable control group [6]. In our cohort of cases, GMF patients tended to show longer OS (132 versus 70 months), perhaps indicating biological differences with MFLCT. In addition, deeper dermal infiltrate in GMF has been reported as not associated with worse prognosis when compared to classic MF, which may further support an innate biological difference in GMF [7]. Our knowledge about biomarkers potentially implicated in the biology of the disease—that could be translated in clinical practice—and how they differ from other aggressive forms of MF is still very limited. GMF shows lower levels of expression of CD30 than MFLCT (4% versus 80%), suggesting higher immune activation in MFLCT (*p* = 0.596). This study, evaluating 28 patients with GMF, is the largest single-institution clinicopathologic case series published thus far, and it is the first to evaluate the status of IC and Th markers in this MF variant.

In agreement with other reported case series, ref. [7,15], GMF patients in our study were predominantly males in their sixth decade, of White ethnicity, and initially presenting with patches, papules, or plaques and at early stage (Stage I/II/III) of disease and with lower levels of LDH than patients with MFLCT. Although initial LDH levels may not be clinically relevant, lower stage at diagnosis may be associated with the fact that LCT usually occurs years after the initial diagnosis. Two patients in our cohort (GMF, n = 1 and MFLCT, n = 1) had a previous medical history of CHL, also reported in other studies, which is an intriguing association since CHL is known to frequently present with granulomatous inflammation [1,7,19,20]. Also consistent with the current literature is that patients with MFLCT have a higher tumor burden when compared with GMF, and for this reason they were significantly more associated with elevated LDH and advanced clinical stage [21].

The pathologic mechanisms of granuloma formation in cases of GMF are still unknown, but Th17 upregulation playing a role is possible, based on our findings, where GMF had a significantly higher expression of RORgT than MFLCT cases. Increased levels of IL-1, IL-6, IL-17, and IL-23 due to the upregulation of this cell population are described to play a central role in granuloma formation and maintenance in other granulomatous diseases of inflammatory or infectious origin, such as sarcoidosis and paracoccidioidomycosis [10,11].

It is widely accepted that early MF tends to show Th1 phenotype, while advanced cases are usually of Th2 phenotype [22]. Based on gene expression profiling [23,24], Amador and colleagues recently assessed Th1 (Tbet and CXCR3) and Th2 (GATA3 and CCR4) protein expression using the IHC algorithm in cases of nodal and extranodal peripheral T-cell lymphoma, not otherwise specified (PTCL, NOS). Cases with a Th2 signature had phosphatidylinositol 3-kinase (PI3K)–mammalian target of rapamycin (mTOR) activation and were related to worse outcome [9]. Despite the fact that Th1 or Th2 phenotype currently does not dictate a particular therapy, the application of this algorithm in clinical practice may facilitate future management and risk stratification. In our cases, GMF showed a higher expression of Tbet, a Th1 marker, possibly indicating a Th landscape similar to early MF. The upregulation of the Th1 axis is also observed in other granulomatous diseases and multisystem inflammatory granulomatous disorders, such as sarcoidosis [25,26]. Interestingly, Th17 has also been implicated in the pathogenesis of cutaneous T-cell lymphomas (CTCLs), being associated with progressive disease [27]. Due to the small number of cases in each of the GMF and MFLCT groups, the evaluation of the variables related to prognosis in each single entity was not possible. However, when combined, RORγT and Foxp3 expression in less than 10% of neoplastic cells was associated with worse prognosis only in the univariate analysis. These findings could be related to many factors, including a different biology of mycosis fungoides compared to other systemic T-cell lymphomas, Th1/Th17 plasticity, and different immune composition of the topography of the lesions [28]. Miyagaki and colleagues have evaluated the cytokine profiling of Th17 and Th22 response in the microenvironment of cutaneous T-cell lymphomas and found an upregulation of IL-22 and downregulation of IL-17A, which could explain the migration of dermal Langerhans cells and the low number of neutrophils, respectively [12]. This downregulation of the Th17 response associated with IL-17 gene polymorphisms, which is also common in this group, could be a possible explanation for the increased rated of bacterial infections [12,29].

Better understanding of the role of the tumor microenvironment is crucial for the discovery of new therapeutic options for patients with MF, especially when long-term disease control is the goal [30]. Although, in general, it seems that the use of currently available IC inhibitors (ICIs) in MF is less promising than in other tumors, it is still accepted that unraveling the complex interaction between MF and microenvironment will lead to improved therapies [31]. Currently, most attempts to treat MF with ICI comprise phase I/II clinical trials, most of them have a limited number of participants. In a recent review, only five of such trials concluded and were published, and most of them involved PD-1 inhibition, and six other trials were still ongoing [30]. The expression of PD-1 and PD-L1 IC in cases of MF is usually evaluated only in special situations for therapeutic purposes, and there are no data in cases of GMF [30,31,32]. Recently, Pileri and colleagues used IHC to assess both markers in neoplastic cells and tumor-associated lymphocytes [33]. The number of positive cells per high-power field in both compartments was overall low and very heterogeneous, but there was a slightly higher positive rate of positive PD-1 in the tumor compartment [33]. A similar overall low expression by tumor cells was observed in our patients. However, these findings do not completely discourage the possibility of ICI. The phase II trial CITN-10 showed a significant antitumor response of pembrolizumab in relapsed/refractory MF, and the treatment response did not correlate with the levels of protein and RNA expression [34]. The antitumor activity of ICI is usually related to toxicity due increased levels of Th17 response [35,36]. Minimal adverse events were observed in this study, but this should be of special interest when tested in GMF due to the baseline increase in Th17, as observed in our evaluation. 

The assessment of Th proteins by IHC can be easily applied in clinical practice, but this assay has some limitations: the limited number of markers used to assess co-expression in a single slide, as well as precise quantification using digital tools of these different combinations of phenotypes. This is especially important for diseases in which the phenotype of the neoplastic cells is defined by a combination of several immune markers, as occurs in MF. We also acknowledge other limitations of the current study, including a retrospective single-center experience, relatively small population of GMF patients, and the inability to use functional and genomic assays due to the limited amount of remaining tissue for testing. 

## 5. Conclusions

In conclusion, proteins associated with the Th1/Th17 axes, along with PD-L1, appear to be upregulated in GMF, findings that seem to be unique among MF subtypes and might be in relation to the characteristic histopathological appearance of this disease. Our findings shed light on the complex immune interaction between MF cells and tumor microenvironment in this rare variant and might open potential therapeutic options for these patients. In larger studies, applying more accurate techniques will help elucidate the precise role of Th and immune checkpoint markers in the pathogenesis of GMF and may drive novel treatment strategies for these patients. 

## Figures and Tables

**Figure 1 cells-13-00419-f001:**
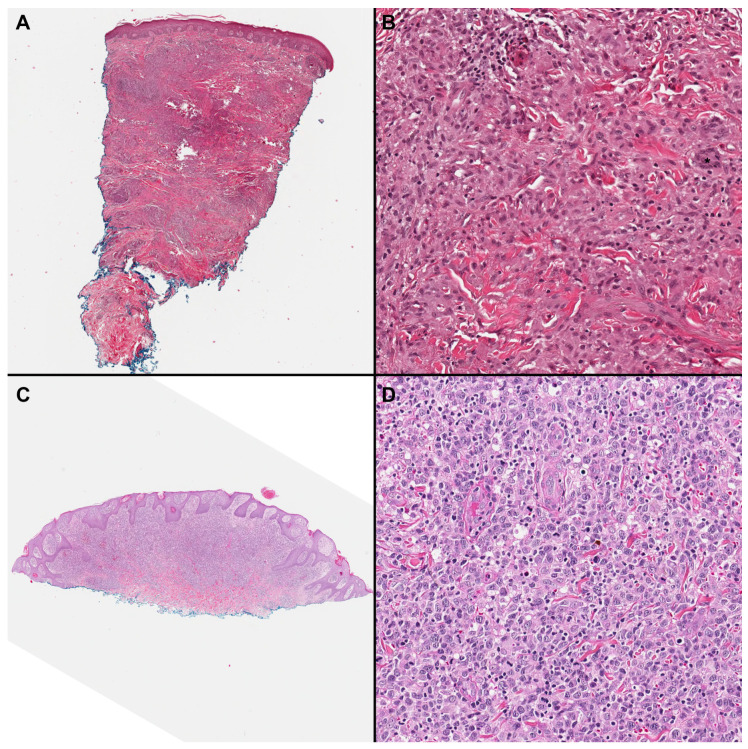
(**A**,**B**) Histopathological appearance of granulomatous mycosis fungoides (GMF). A dense dermal infiltrate without marked epidermotropism and composed of atypical small to medium lymphocytes is seen in association with a prominent granulomatous infiltrate with multinucleated giant cells (*) (H&E, 5×, 200×); (**C**,**D**) A case of mycosis fungoides with large cell transformation (MFLCT) shows an atypical infiltrate occupying dermis and showing focal epidermotropism. The infiltrate is predominantly composed of atypical medium and large lymphocytes (H&E, 5×, 200×).

**Figure 2 cells-13-00419-f002:**
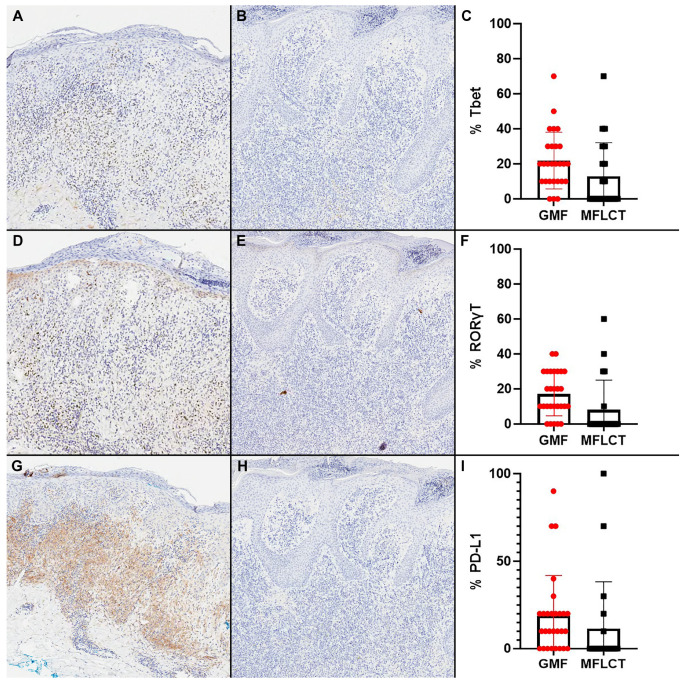
Characterization of T helper (Th) markers in cases of granulomatous mycosis fungoides (GMF) and mycosis fungoides with large cell transformation (MFLCT). (**A**–**C**) GMF (**A**) shows an upregulation of Tbet expression (**C**) when compared to MFLCT (**B**) (immunohistochemistry, 20×, 50×); (**D**–**F**) GMF (**D**) shows an upregulation of RORγT expression (**F**) when compared to MFLCT (**E**) (immunohistochemistry, 20×, 50×); (**G**–**I**) GMF (**G**) shows an upregulation of PD-L1 expression in tumor and inflammatory cells (**I**) when compared to MFLCT (**H**) (immunohistochemistry, 20×, 50×).

**Figure 3 cells-13-00419-f003:**
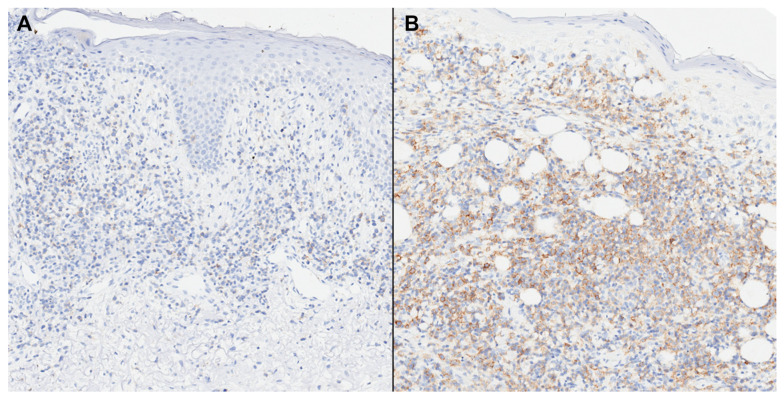
PD-1 expression in granulomatous mycosis fungoides (GMF) (**A**) and mycosis fungoides with large cell transformation (MFLCT) (**B**). There is strong (3+) labeling for PD-1 in MFCLT in contrast to most cases of GMF (immunohistochemistry, 50×).

**Table 1 cells-13-00419-t001:** Cohort baseline characteristics.

	Cohort, n = 49 (%)	GMF, n = 28 (%)	MFLCT, n = 21 (%)	*p*-Value
Age (yrs)				0.067
Median	60	58	61	
Range	21–77	21–67	34–77	
Sex				0.356
Male	29 (59)	15 (54)	14 (67)	
Female	20 (41)	13 (46)	7 (33)	
Ethnicity				0.540
White	35 (71)	19 (68)	16 (76)	
Black	11 (23)	8 (28)	3 (14)	
Hispanic	2 (4)	1 (4)	1 (5)	
Unknown	1 (2)	0 (0)	1 (5)	
Previous history of malignancy	11 (23)	6 (21)	5 (23)	0.843
History of previous treatments	46 (93)	25 (89)	21 (100)	0.250
LDH, median (range), IU/L	543 (128–1035)	501 (128–836)	583 (338–1035)	0.042
β2M, median (range), mcg/mL	2.7 (1.6–4.8)	2.45 (2.4–4.8)	3.15 (1.6–4.5)	0.828
Clinical presentation				0.897
Papules	5 (10)	2 (7)	3 (14)	>0.999
Patch	34 (69)	18 (64)	16 (76)	0.665
Plaques	37 (75)	19 (67)	18 (25)	0.969
Tumor	16 (32)	7 (25)	9 (42)	0.582
Clinical Stage at Diagnosis				0.036
I	13 (26)	11 (39)	1 (5)	
II	16 (32)	8 (29)	8 (38)	
III	0 (0)	0 (0)	0 (0)	
IV	13 (26)	4 (14)	4 (19)	
Unknown	7 (16)	5 (18)	8 (38)	
Status at last follow-up				N/A
ANED	2 (4)	1 (3)	1 (4)	
AWD	17 (34)	15 (53)	2 (8)	
DOD	30 (62)	12 (44)	18 (88)	

N/A: Nonapplicable; β2M: beta-2-microglobulin; ANED: alive with no evidence of disease; AWD: alive with disease; DOD: died of disease; GMF: granulomatous mycosis fungoides; LDH: lactate dehydrogenase; MFLCT: mycosis fungoides with large cell transformation; yrs: years.

**Table 2 cells-13-00419-t002:** Immunohistochemical markers and their expression used in the diagnostic assessment of patients with granulomatous mycosis fungoides (GMF) and mycosis fungoides with large cell transformation (MFLCT).

Antibody	Number of Positive Cases			
	Cohort (%)	GMF (%)	MFLCT (%)	*p*-Value
CD1a	0/1 (0)	0/1 (0)	N/A	N/A
CD2	N/A	N/A	N/A	N/A
CD3	32/32 (100)	24/24 (100)	8/8 (100)	>0.999
CD4	29/31 (93)	22/23 (95)	7/8 (87.5)	0.455
CD5	N/A	N/A	N/A	N/A
CD7	6/22 (27)	4/19 (21)	2/3 (66)	0.168
CD8 *	6/30 (25)	3/23 (13)	3/7 (42)	0.120
CD20	0/7 (0)	0/5 (0)	0/2 (0)	>0.999
CD25	12/25 (48)	3/8 (37.5)	9/17 (53)	0.672
CD30	37/40 (92)	17/19 (89)	20/21 (95)	0.596
Median% (range)	50 (1–100)	4 (1–50)	80 (50–100)	<0.0001
EBER	0/3 (0)	0/2 (0)	0/1 (0)	>0.999
TCRBF1	3/3 (100)	3/3 (100)	N/A	N/A
TCRG	0/3 (0)	0/3 (0)	N/A	N/A

N/A: Not available or applicable; * CD8 was considered positive in cases with a shift in the normal CD4/CD8.

**Table 3 cells-13-00419-t003:** Immunohistochemical expression of T helper and immune checkpoint markers in patients with granulomatous mycosis fungoides (GMF) and mycosis fungoides with large cell transformation (MFLCT).

	Cohort	GMF	MFLCT	
Tbet (%)				0.017
Mean (SD)	18 (0.17)	22 (0.16)	13 (0.19)	
Median	15	20	0	
Intensity, n				0.252
1+	1	0	1	
2+	4	3	1	
3+	28	21	7	
GATA3 (%)				0.422
Mean (SD)	42 (0.29)	40 (0.26)	46 (0.33)	
Median	40	40	40	
Intensity, n				0.453
1+	4	1	3	
2+	7	4	3	
3+	31	18	13	
RORγT (%)				0.001
Mean (SD)	13 (0.15)	17 (0.12)	8 (0.16)	
Median	10	15	0	
Intensity, n				0.518
1+	6	4	2	
2+	9	8	1	
3+	13	11	2	
Foxp3 (%)				0.112
Mean (SD)	15 (0.13)	17 (0.12)	13 (0.15)	
Median	10	15	10	
Intensity, n				0.113
1+	1	1	0	
2+	2	0	2	
3+	34	23	11	
PD-1 (%)				0.589
Mean (SD)	22 (0.26)	22 (0.25)	21 (0.29)	
Median	10	10	10	
Intensity, n				0.031
1+	0	0	0	
2+	6	6	0	
3+	23	12	11	
PD-L1 (%)				0.011
Mean (SD)	16 (0.24)	19 (0.22)	12 (0.26)	
Median	10	10	0	
Intensity, n				0.814
1+	0	0	0	
2+	6	5	1	
3+	19	15	4	

GMF: granulomatous mycosis fungoides; MFLCT: mycosis fungoides with large cell transformation.

**Table 4 cells-13-00419-t004:** Univariate analyses of the clinicopathologic variables.

Variables	Univariate Analysis (*p* Value)
**Sex**	**0.547**
**Ethnicity**	**0.880**
**Previous History of malignancy**	**0.371**
**LDH**	**0.0555**
**Β2 microglobulin**	**0.071**
**%Tbet (<15% vs. ≥ 15%)**	**0.410**
**%GATA3 (<40% vs. ≥ 40%)**	**0.675**
**%PD-1 (<10% vs. ≥ 10%)**	**0.562**
**%PD-L1 (<10% vs. ≥ 10%)**	**0.787**
**Age**	** *0.017* **
**Clinical Stage at Diagnosis**	** *<0.001* **
**Group (GMF vs. MFLCT)**	** *0.018* **
**%RORγT (<10% vs. ≥ 10%)**	** *0.006* **
**%Foxp3 (<10% vs. ≥ 10%)**	** *0.002* **

## Data Availability

The datasets used and/or analyzed during the current study are available from the corresponding author on reasonable request.

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
