# Peer review of "Differential Upregulation of Th1/Th17-Associated Proteins and PD-L1 in Granulomatous Mycosis Fungoides"

_cells, 2024, doi:10.3390/cells13050419_

Round 1

Reviewer 1 Report

Comments and Suggestions for Authors

The authors present a retrospective study on a rarer subtype of MF, i.e. the granulomatous type (GMF). In doing so, they aim at comparing the clinical and histological of such subtype with the more common and aggressive form of MF, known as the transformed type (TMF). Whilst the descriptive findings may be interesting, as data on GMF are lacking, the inferential approach used in this paper is flawed by major limitations. For instance:

1. It is not clear the reason why the authors decide to compare GMF with TMF a priori. The latter entity can be the result of several types of MF. Such choice may seem as a quite random choice (for instance, why not to compare it with patch-stage MF? Tumor-stage MF? Sezary patients?). If the authors foresee some kind of biological rationale behind this choice, this must be stated. 

2. The authors state that "Expression of CD30 was frequent (17/19, 89%, median expression: 4%) in GMF group". How was such feature distinguished from transformed CD30+ MF? Were there any cases of transformed GMF?

3. The authors state that "GMF group had a statistically significant association with lower levels of LDH at 160 clinical presentation (median, 501 vs 583 IU/L) (p= 0.042) and lower clinical stage (Stage I) 161 at the time of the diagnosis (p= 0.036) when compared to MFLCT". The authors should comment on these findings. The former seems clinical irrelevant, the latter may seem obvious, as large-cell transformation usually occurs after some years from the diagnosis of MF. 

4. Page 4: please correct the typo "OS was 1216 months" (126?) 

5. Advanced clinical stage as a poor predictor of survival is a well known clinical feature. Please define it for those readers not experienced in the field (I believe the authors mean stage IIB-IV). 

6. How many variables were considered in the multivariate analysis? Two at the time? All 5 at the same time? It is not clear. It would be apt to show the number of events (i.e. deaths) to double-check such analysis for the sample size. 

6. The discussion needs to be improved. What are the possibile implications of these findings? Recently, two interesting articles have dealt with the role of microenvironment (DOI: 10.3390/cancers15030746) and PD1 axis in MF (DOI: 10.3389/fonc.2021.733770). I would suggest to make a comment on such findings and their potential implications. 

Author Response

January 10, 2023

Dr. Silvia Alberti-Violetti and Dr. Gianluca Avallone

Cells Editors “New Discoveries in Dermatopathology: From Molecular Mechanisms to Therapeutic Opportunities"

REF: Resubmission of manuscript “Differential Upregulation of Th1/Th17-associated proteins and PD-L1 in Granulomatous Mycosis Fungoides”

Dear Drs. Alberti-Violetti and Avallone:

Thank you for considering our article for publication and for the valuable comments and suggestions from the reviewers. Please find below our detailed answers to the reviewers’ comments. We are convinced that our article has greatly improved thank to the reviewers’ suggestions, which we have all included in the revised version.

Please let me know if you have any further questions.

Sincerely,

Carlos A. Torres-Cabala, MD (corresponding autor)

Professor

Departments of Pathology, Dermatopathology, and Translational Molecular Pathology

Chief, Dermatopathology Section

The University of Texas MD Anderson Cancer Center

Reviewer #1

The authors present a retrospective study on a rarer subtype of MF, i.e. the granulomatous type (GMF). In doing so, they aim at comparing the clinical and histological of such subtype with the more common and aggressive form of MF, known as the transformed type (TMF). Whilst the descriptive findings may be interesting, as data on GMF are lacking, the inferential approach used in this paper is flawed by major limitations. For instance:

  1. It is not clear the reason why the authors decide to compare GMF with TMF a priori. The latter entity can be the result of several types of MF. Such choice may seem as a quite random choice (for instance, why not to compare it with patch-stage MF? Tumor-stage MF? Sezary patients?). If the authors foresee some kind of biological rationale behind this choice, this must be stated. 

We thank the reviewer for this very important comment. Although we completely agree that selecting TMF cases for comparison may appear arbitrary, our rationale was based on reported clinical and biological features of GMF. Per Li et al (J Am Acad Dermatol 2013;69:366-74), “granulomatous involvement in MF represents a prognostic marker for worsening disease… disease progression was seen in 46% of patients with GMF compared with 30% of control patients… The 5- and 10-year PFS were significantly worse in the GMF group…”. PFS was 59% (5 years) and 33% (10 years) for GMF compared to 84% (5 years) and 56% (10 years) for classic MF. TMF has been reported to have PFS of 45% (5 years) and 22% (10 years) (Talpur et al, Clin Lymph Myel Leuk 2016;16:49-56), thus putting GMF closer to TMF than to regular MF in terms of prognosis. In addition, “the deeper dermal infiltrate in GMF was not associated with worse prognosis when compared with patients with classic MF and may reflect the biologic differences of GMF” (Li et al), adding a potential biological rationale that may explain why granulomatous features per se are associated with a worse prognosis. It seems to us, therefore, that comparing GMF with cases of MF that show similar clinical behaviour (TMF) is appropriate, although we acknowledge the reviewer’s argument that it may not be the perfect comparison. We have added a sentence explaining our rationale in the Introduction, another expanding this in the Discussion, and added the Talpur et al reference, as follows:

Introduction: “Disease progression in GMF has been reported as high as 46%, in contrast to 30% in classic MF, and comparable to that reported for MF with LCT.”

Discussion: “Progression free survival (PFS) of GMF has been reported as 59% and 33% at 5 and 10 years, compared to 84% and 56% for classic MF (15). PFS of GMF is closer to that shown by MFLCT (45% and 22%) (Talpur et al), making MFLCT a reasonable control group. In our cohort of cases, GMF patients tended to show longer OS (132 versus 70 months, no difference found in MVA), perhaps indicating some biological differences with MFLCT. In addition, deeper dermal infiltrate in GMF has been reported as not associated with worse prognosis when compared to classic MF, which may further support innate biological difference in GMF (15).”

  1. The authors state that "Expression of CD30 was frequent (17/19, 89%, median expression: 4%) in GMF group". How was such feature distinguished from transformed CD30+ MF? Were there any cases of transformed GMF?

We thank the reviewer for this insightful observation. In Table 2 we show that transformed MF cases also frequently expressed CD30 (20/21, 95%), with no difference between the two groups (p=0.596) when numbers of positive (>1%) cases were compared. However, Transformed MF cases had a higher median percentage of expression (80% versus 4%). This is stated in section 3.4 Comparison of baseline characteristics between GMF and MFLCT. Only two of the GMF cases had features of large cell transformation; this is stated in section 3.2, page 4, line 154.

Differences in CD30 level of expression (4% versus 80%) are now discussed as follows:

Discussion: “GMF shows lower levels of expression of CD30 than MFLCT (4% versus 80%), suggesting higher immune activation in MFLCT.”

  1. The authors state that "GMF group had a statistically significant association with lower levels of LDH at clinical presentation (median, 501 vs 583 IU/L) (p= 0.042) and lower clinical stage (Stage I) at the time of the diagnosis (p= 0.036) when compared to MFLCT". The authors should comment on these findings. The former seems clinical irrelevant, the latter may seem obvious, as large-cell transformation usually occurs after some years from the diagnosis of MF. 

We thank the reviewer for this comment. We added a sentence in the discussion section about these findings.

Discussion: “GMF patients in our study… initially presenting with patches, papules or plaques and at early stage (Stage I/II) of the disease and with lower levels of LDH than patients with MFLCT. Although initial LDH levels may not be clinically relevant, lower stage at diagnosis may be associated with the fact that LCT usually occurs after years from the initial diagnosis.”

  1. Page 4: please correct the typo "OS was 1216 months" (126?) 

We thank the Reviewer for this observation. We have corrected the typo and the OS should be 121.6 months.

  1. Advanced clinical stage as a poor predictor of survival is a well-known clinical feature. Please define it for those readers not experienced in the field (I believe the authors mean stage IIB-IV). 

We appreciate the Reviewer’s comment. We indeed referred to patients with stage IIB to IV as advanced. In order to be more precise we have replaced “advanced stage” by the actual stage (mostly stage IV) in the revised manuscript.

  1. How many variables were considered in the multivariate analysis? Two at the time? All 5 at the same time? It is not clear. It would be apt to show the number of events (i.e. deaths) to double-check such analysis for the sample size. 

We thank the Reviewer for the question. For the multivariate analysis, we considered all 5 statistically significant variables of the univariate analysis at the same time.

The number of events of the univariate analysis for each of the categorical variable are respectively: clinical stage (Stage 1: 5 events in 13 patients; Stage 2: 11 events in 16 patients; Stage 4: 12 in 13 patients), group (GMF: 12 events in 28 patients; MFLCT: 18 events in 21 patients), %RORgT (<10%: 17 events in 21 patients; ≥10%: 13 events in 28 patients), and %Foxp3 (<10%: 10 events in 12 patients; ≥10%: 20 events in 37 patients). We have composed a Table with the number of events and added it to the Supplemental materials.

  1. 7. The discussion needs to be improved. What are the possibile implications of these findings? Recently, two interesting articles have dealt with the role of microenvironment (DOI: 3390/cancers15030746) and PD1 axis in MF (DOI: 10.3389/fonc.2021.733770). I would suggest to make a comment on such findings and their potential implications. 

We thank the Reviewer for this suggestion and the previous comments, that have improved our Discussion substantially. We added both references in the revised discussion and included a comment about the potential implications of our findings regarding the role of tumor microenvironment. Our revised conclusion emphasizes that our findings contribute to the better understanding of tumor microenvironment in MF, especially the rare GMF, with potential impact in therapy.

Discussion: “Better understanding of the role of the tumor microenvironment is crucial for the discovery of new therapeutic options for patients with MF, especially when long-term disease control is the goal (Dobos et al). Although, in general, it seems that the use of currently available IC inhibitors (ICI) in MF is less promising than in other tumors, it is still accepted that unraveling the complex interaction between MF and microenvironment will lead to improved therapies (Roccuzzo et al). Currently, most attempts to treat MF with ICI comprise phase I/II clinical trials, most of them with a limited number of participants. In a recent review, only five of such trials concluded and were published, most of them involved PD-1 inhibition, and 6 other trials were still ongoing (Roccuzzo et al).”  

Conclusion: “Our findings shed light on the complex immune interaction between MF cells and tumor microenvironment in this rare variant, and might open potential therapeutic options for these patients.”

Reviewer 2 Report

Comments and Suggestions for Authors

The manuscript of Marques-Piubelli and Co-workers does represent an example of clinical-immunohistochemical study, focusing on a rare form of cutaneous peripheral T-cell lymphoma: granulomatous mycosis fungoides (GMF). Twenty-eight cases of GMF were evaluated and compared with 21 examples of MF with large cell transformation (MFLCT). The Authors honestly admit the limitations of their study, including the relatively small panel of markers applied and the lack of use of the most recent molecular techniques, such as single cell spatial transcriptomics, which can actually highlight the relationships between tumoral and microenvironmental cells. Concerning the latter, it would be wise to assess the number of epithelioid elements expressing PD-L1. In fact, looking at figure 2G one has the impression that cells other than the neoplastic ones turn positive for PD-L1.

Author Response

January 10, 2023

Dr. Silvia Alberti-Violetti and Dr. Gianluca Avallone

Cells Editors “New Discoveries in Dermatopathology: From Molecular Mechanisms to Therapeutic Opportunities"

REF: Resubmission of manuscript “Differential Upregulation of Th1/Th17-associated proteins and PD-L1 in Granulomatous Mycosis Fungoides”

Dear Drs. Alberti-Violetti and Avallone:

Thank you for considering our article for publication and for the valuable comments and suggestions from the reviewers. Please find below our detailed answers to the reviewers’ comments. We are convinced that our article has greatly improved thank to the reviewers’ suggestions, which we have all included in the revised version.

Please let me know if you have any further questions.

Sincerely,

Carlos A. Torres-Cabala, MD (corresponding autor)

Professor

Departments of Pathology, Dermatopathology, and Translational Molecular Pathology

Chief, Dermatopathology Section

The University of Texas MD Anderson Cancer Center

Reviewer #2

The manuscript of Marques-Piubelli and Co-workers does represent an example of clinical-immunohistochemical study, focusing on a rare form of cutaneous peripheral T-cell lymphoma: granulomatous mycosis fungoides (GMF). Twenty-eight cases of GMF were evaluated and compared with 21 examples of MF with large cell transformation (MFLCT). The Authors honestly admit the limitations of their study, including the relatively small panel of markers applied and the lack of use of the most recent molecular techniques, such as single cell spatial transcriptomics, which can actually highlight the relationships between tumoral and microenvironmental cells. Concerning the latter, it would be wise to assess the number of epithelioid elements expressing PD-L1. In fact, looking at figure 2G one has the impression that cells other than the neoplastic ones turn positive for PD-L1.

We thank Reviewer #2 for the insightful comment and suggestion. Evaluation of PD-L1 is difficult when inflammatory infiltrate is present, like in our cases. Determining the actual number of non-tumor (epithelioid) cells in our cases would not be possible without using sophisticated techniques like multiplex immunofluorescence, which we did not apply. Following current guidelines, we considered PD-L1 positivity only when membranous labelling was present; however, cytoplasmic expression is also known to occur.

We completely agree that the associated inflammatory cells composing the tumor microenvironment (including epithelioid histiocytes) were positive, as shown in figure 2G, and we have modified the Figure legend accordingly to acknowledge this finding.

Reviewer 3 Report

Comments and Suggestions for Authors

The authors compared the clinicopathological features and the expression of PD-1, PD-L1, and master genes of each T-cell lineages in lesional skin assessed by immunohistochemistry in granulomatous mycosis fungoides (GMF) patients to those in patients with MF with large cell transformation (MF-LCT). They found that GMF patients were associated with early clinical stage and showed higher percentage of cells positive for Tbet, RORγT, and PD-L1. On the other hand, PD-1 intensity was stronger in patients with MF-LCT. In multivariate analysis, these parameters were not associated with the prognosis and the advanced stage was the only factor related to a poor prognosis in the authors’ cohorts. This is a well-written paper and I have read the manuscript with great interest. I have some concerns below.

1.     It will be better to add the histopathological images with higher magnification to clearly show the granulomatous changes.

2.     I am interested in whether Tbet, RORγT, and PD-L1 expression levels are different between GMF and classical MF without LCT. There is a possibility that higher clinical stages in MF-LCT patients may cause the differences between GMF and MF-LCT in this report.

3.     Concerning Th17 involvement in MF, some reports suggested that IL-17 was associated with the progression of MF as the reference authors cited in the discussion. However, other reports showed the low expression levels of IL-17 in blood or skin of MF patients (Miyagaki T et al., Clin Cancer Res 2011; 17: 7529-38; Kolkowsky K et al., Acta Derm Venereol 2022; 102: adv00777). Probably Th17 involvement may be different depending on individual cases in classical MF. It might be the different characteristic of classical MF from GMF. Anyway, Th17 involvement in MF should be summarized and added in the Introduction or Discussion section.

Author Response

January 10, 2023

Dr. Silvia Alberti-Violetti and Dr. Gianluca Avallone

Cells Editors “New Discoveries in Dermatopathology: From Molecular Mechanisms to Therapeutic Opportunities"

REF: Resubmission of manuscript “Differential Upregulation of Th1/Th17-associated proteins and PD-L1 in Granulomatous Mycosis Fungoides”

Dear Drs. Alberti-Violetti and Avallone:

Thank you for considering our article for publication and for the valuable comments and suggestions from the reviewers. Please find below our detailed answers to the reviewers’ comments. We are convinced that our article has greatly improved thank to the reviewers’ suggestions, which we have all included in the revised version.

Please let me know if you have any further questions.

Sincerely,

Carlos A. Torres-Cabala, MD (corresponding autor)

Professor

Departments of Pathology, Dermatopathology, and Translational Molecular Pathology

Chief, Dermatopathology Section

The University of Texas MD Anderson Cancer Center

Reviewer #3

The authors compared the clinicopathological features and the expression of PD-1, PD-L1, and master genes of each T-cell lineages in lesional skin assessed by immunohistochemistry in granulomatous mycosis fungoides (GMF) patients to those in patients with MF with large cell transformation (MF-LCT). They found that GMF patients were associated with early clinical stage and showed higher percentage of cells positive for Tbet, RORγT, and PD-L1. On the other hand, PD-1 intensity was stronger in patients with MF-LCT. In multivariate analysis, these parameters were not associated with the prognosis and the advanced stage was the only factor related to a poor prognosis in the authors’ cohorts. This is a well-written paper and I have read the manuscript with great interest. I have some concerns below.

  1. It will be better to add the histopathological images with higher magnification to clearly show the granulomatous changes.

We thank the Reviewer for the suggestion. We have modified Figure 1 and the revised version is a higher magnification image to better show the granulomatous changes. 

  1. I am interested in whether Tbet, RORγT, and PD-L1 expression levels are different between GMF and classical MF without LCT. There is a possibility that higher clinical stages in MF-LCT patients may cause the differences between GMF and MF-LCT in this report.

This is a very fair statement. We chose to use MFLCT as a control group since GMF has been reported to have a prognosis comparable to that of MFLCT and better than that of MF without LCT. Expanding our study to patients with MF without LCT is definitely the next step in this research project. We assessed the percentage of expression of the markers in different clinical stages of the whole cohort using ANOVA, as per the Reviewer’s suggestion, and we found the following: Tbet [p= 0.7983, median expression: Stage 1(20%), Stage 2 (10%), and Stage 4 (10%)], RORγT [p= 0.1399, median expression: Stage 1(10%), Stage 2 (15%), and Stage 4 (0%)], and PD-L1[p= 0.5838, median expression: Stage 1(10%), Stage 2 (10%), and Stage 4 (0%)], thus no association between staging and expression of Tbet, RORgT or PD-L1 was identified.

We have added a sentence in section 3.5: “No association between staging and expression of Tbet, RORgT or PD-L1 was identified.”  

  1. Concerning Th17 involvement in MF, some reports suggested that IL-17 was associated with the progression of MF as the reference authors cited in the discussion. However, other reports showed the low expression levels of IL-17 in blood or skin of MF patients (Miyagaki T et al., Clin Cancer Res 2011; 17: 7529-38; Kolkowsky K et al., Acta Derm Venereol 2022; 102: adv00777). Probably Th17 involvement may be different depending on individual cases in classical MF. It might be the different characteristic of classical MF from GMF. Anyway, Th17 involvement in MF should be summarized and added in the Introduction or Discussion section.

We thank the Reviewer for this important comment. In our revised manuscript, sentences about expression of IL-17 in MF progression have been added in the introduction and discussion sections, along with the cited reference.

Introduction: “The cytokine profiling of Th17 response of the immune microenvironment in cutaneous T-cell lymphomas have been assessed and it seems to be dysregulated, promoting an imbalance of the neutrophil component (Miyagaki et al)”

Discussion: “Miyagaki and colleagues have evaluated the cytokine profiling of Th17 and Th22 response in the microenvironment of cutaneous T-cell lymphomas and found an upregulation of IL-22 and downregulation of IL-17A, which could explain migration of dermal Langerhans cells and the low number of neutrophils, respectively [10]. This downregulation of the Th17 response associated to IL-17 gene polymorphisms, which is also common in this group, could be a possible explanation of the increased rated of bacterial infections.”

Round 2

Reviewer 1 Report

Comments and Suggestions for Authors

Thank you for the extensive revisions. Still, it is essential to enhance the multivariate analysis. The authors have indicated a total of 30 events in the supplementary table related to groups, ROR, and FOX. However, the clinical stage section mentions only 28 events. Please verify this inconsistency. Additionally, analyzing five variables simultaneously in the multivariate analysis is incorrect. If the dataset comprises only 30 events, a maximum of 2 or 3 variables should be analyzed concurrently to avoid overfitting. Kindly reconsider the analysis accordingly.

Author Response

Reviewer #1
Thank you for the extensive revisions. Still, it is essential to enhance the multivariate 
analysis. The authors have indicated a total of 30 events in the supplementary table 
related to groups, ROR, and FOX. However, the clinical stage section mentions only 28 
events. Please verify this inconsistency. Additionally, analyzing five variables 
simultaneously in the multivariate analysis is incorrect. If the dataset comprises only 30 
events, a maximum of 2 or 3 variables should be analyzed concurrently to avoid 
overfitting. Kindly reconsider the analysis accordingly.
We thank the Reviewer #1 for the comments. There were 7 patients in the cohort in 
which the clinical stage was unknown and two of them had events. We added it to the 
new supplementary table. Regarding the MVA, we thank the reviewer for this comment. 
For this reason, we left clinical stage, %RORgT, and %Foxp3 (the 3 variables with 
stronger correlation in univariate analysis) for the multivariate analysis. Based on that, 
clinical stage is still the only variable significant in MVA. 

Reviewer 3 Report

Comments and Suggestions for Authors

The authors’ responses are convincing and no more concerns are remained.

Author Response

Reviewer #3
The authors’ responses are convincing and no more concerns are remained.
We thank the Reviewer #3 for the comment.